# Green Contributions to the Chemistry of Perezone and Oxidation of the Double Bond of the Side Chain: A Theoretical Study and Cytotoxic Evaluation in MDA-MB231 Cells

**DOI:** 10.3390/molecules30234603

**Published:** 2025-11-30

**Authors:** René Gerardo Escobedo-González, Joel Martínez, Adriana L. Rivera-Espejel, Claudia L. Vargas-Requena, María Inés Nicolás-Vázquez, René Miranda Ruvalcaba

**Affiliations:** 1Department of Industrial Maintenance and Nanotechnology, Technological University of Juarez City, City of Juarez 32695, Chihuahua State, Mexico; 2Departamento de Ciencias Químicas, Facultad de Estudios Superiores Cuautitlán Campo 1, Universidad Nacional Autónoma de México, Cuautitlán Izcalli 54740, State of Mexico, Mexico; atlanta126@gmail.com (J.M.); riveraespejeladriana@gmail.com (A.L.R.-E.); nicovain@yahoo.com.mx (M.I.N.-V.); 3Institute of Biomedical Science, Autonomous University of the City of Juarez (UACJ), City of Juarez 32315, Chihuahua State, Mexico; cvargas@uacj.mx

**Keywords:** green approach, perezone-isoperezone, DFT study, antineoplastic compounds, molecular docking, chemoinformatic study

## Abstract

Perezone, a sesquiterpene quinone, was the first natural product isolated in crystalline form on the American continent in 1852. It is commonly found in the roots of herbs from the Acourtia species (formerly *Perezia*). This molecule, along with its synthetic isomer isoperezone, exhibits antineoplastic effects, among others. In this study, an enzymatic reaction (green chemistry) was employed to oxidize the C12−C13 double bond of perezone and isoperezone. This method proved to be more effective than traditional toxic chemical oxidants. As result, epoxides were obtained, followed by acetonides, diols, and esters. All compounds were successfully synthesized and characterized using standard spectroscopic techniques. In breast cancer cell tests, the isoperezone acetonide showed the highest cytotoxicity, with an IC_50_ of 8.44 µM. Additionally, a computational study was performed at the DFT (B3LYP) level of theory, indicating that the geometrical and energy differences between **6**-*R* and **6**-*S* stereoisomers are 0.5 kcal/mol, and the spectroscopic and electronic properties aligned with the experimental data. Finally, molecular docking revealed binding energies of −8.14 kcal/mol for **6**-*R* and −8.04 kcal/mol for **6**-*S*, with a hydrogen bond of 2.9 Å involving the His121 residue. A chemoinformatic prediction was also conducted to compare cytotoxicity results.

## 1. Introduction

Perezone, pipitzahoic acid, or (*R*)-3-hydroxy-5-methyl-2-(6-methylhept-5-en-2-yl) cyclohexa-2,5-diene-1,4-dione (**1**, Figure 1), is an abundant sesquiterpene quinone in the roots of herbal members of the genus *Acourtia* (formerly *Perezia*) [1]. For many years, perezone has been the focus of several chemical, structural, spectroscopic, pharmacologic, and in silico studies, as recently discussed by our research group [2].

It is important to note that **1** and its synthetic isomer, isoperezone (**2**), are molecules of high interest due to their pharmacological properties; both molecules exhibit coupling with caspases, molecules responsible for programmed cell death [2]. However, chemo-computer studies have indicated that these quinones display a toxicological risk of irritation, attributed to the presence of the double bond of the side chain. Consequently, their chemical modification could modify toxicological risk.

Considering the above commentaries, the goal of this work was, in the first stage, to contribute to the Green Chemistry Protocol (GChP) [3,4] by performing an eco-oxidation of the double bond of the side chain present in **1** and **2** to obtain the corresponding epoxides **3** and **4,** using a biocatalytic reaction with Novozyme^®^ 435 (Principle #9) in the presence of ethyl acetate, a green solvent (Principle #5), at room temperature and atmospheric pressure (Principle #6); then, the production of the respective diols was evaluated using eco-friendly reagents rather than toxic chemical oxidants (Figure 1). In the second stage, a set of computational studies was performed and evaluated which, in addition to acquiring the antineoplastic properties of the obtained products, established an adequate explanation of their activity through the computational analysis results. Finally, an appropriate molecular docking evaluation and a chemoinformatic prediction were accomplished.

## 2. Results and Discussion

### 2.1. Classic Oxidations for the Double Bond of the Side Chain in ***1*** and ***2***

This study was initiated using Bayer reagent (KMnO_4_aq 5%) to generate the corresponding diols of perezone and isoperezone by double bond oxidation. In general, compounds **9** and **10** were produced with low yields (5%). Then, OsO_4_ was also evaluated; the results were unsatisfactory (≤20%) [5]. Finally, another attempt to oxidize the double bond using ozonolysis was performed; the results were undesirable; an excessive number of products was detected by TLC.

### 2.2. Green Oxidations for the Double Bond of the Side Chain in ***1*** and ***2***

Concerning the GChP, the oxidation of the double bond was improved using environmentally friendly reactions (Figure 1). In this sense, a green approach was offered, according to the following advantages: urea and water were generated as green residues (Principle #1); high economy was achieved (Principle #2); the substrate and reagents were environmentally friendly (Principle #3); ethyl acetate, acetone, or water was employed as a green solvent (Principle #5); to promote diols **9** and **10,** microwave irradiation was used, and the other reactions were performed at room temperature and atmospheric pressure (Principle #6); the initial substrate was renewable (Principle #7); Novozyme™ 435 or TAFF™ (Tonsil Actisil FF, a bentonitic clay) [6] was employed as a catalyst (Principle #9); all the reactions were monitored in real time (Principle #11); and the substances used minimize the potential of chemical risk (Principle #12). Consequently, using a relevant green metric [7], this greener evaluation could be considered as a good green approach.

Thus, in the first step, the new epoxides of **1** and **2** were obtained using lipase B from *Candida antarctica* (Novozyme™ 435) as a catalyst, with urea hydroperoxide as the oxidant in ethyl acetate as the solvent at room temperature [8] (Figure 1). In this regard, the target compounds **3** and **4** were produced as amorphous yellow solids with melting points between 78 and 80 °C and 84 and 86 °C, respectively, with good yields of 85%. Additionally, it is important to note that all compounds were obtained as diastereomeric mixtures.

In the second step, epoxides **3** and **4** were catalytically reacted with TAFF™ using anhydrous acetone, a green solvent, at room temperature, producing the new acetonides **5** and **6** (Figure 1), with yields of 50% and 77%, respectively. Both were obtained as yellow amorphous solids, with melting points of 74–75 °C and 102–103 °C, respectively. It is important to note that TAFF™ has both Lewis and Brönsted–Lowry acid features [9]. In this sense, the aluminum–silicate sites of the TAFF™ activate the carbonyl group of the ketone by an acid–base reaction, generating the acetonides [9].

Thus, to avoid the formation of **5** and **6**, the next strategy using TAFF™ with anhydrous ethyl acetate (Figure 1), a green solvent, was performed. Nevertheless, one hydroxyl group was produced, and the other oxygen atom accomplished a transesterification reaction with the solvent, producing the monoesters **7** and **8** in low yields of 34% and 28%, respectively. The target **7** was offered as red oil, and **8** appeared as an amorphous yellow solid with a melting point of 110–112 °C. Hence, the Brönsted–Lowry acid sites of TAFF™ [10] supported a bimolecular nucleophilic substitution with a tetrahedral intermediate [11].

Finally, microwave irradiation as the activating reaction mode was employed to produce **9** and **10**, both with a 50% yield. In this sense, **9** was a deep purple oil, and **10** was obtained as a yellow amorphous solid, with a melting point of 112–114 °C. It is important to note that the acid hydrolysis of epoxides **3** and **4** using water–acetic acid (2:1) ensued with moderate yields (40%).

### 2.3. Structural Attribution

For the structural attribution of the obtained molecules, **3** is offered as a representative example: it is a yellow solid displaying a mp of 78–80 °C. The corresponding mass spectrum (EI-MS) shows a peak *m*/*z* 264 related to the molecular ion; it was validated by HRMS data, offering an elemental composition of C_15_H_20_O_4_ correlated with an exact value of 264.1364 Da, a precise value of 264.1355 Da, and an error of −2.9 ppm, providing an unsaturation data value of 6.0, agreeing with the structure. Related to the ^1^H NMR, it is important to comment that, in correlation with **1** and as expected, an important change occurs only with H12 [2], a signal appreciated as a triplet at *δ*2.71 ppm, denoting a displacement to low frequency due to the electron donor influence of the oxygen atom. Associated with the ^13^C NMR, the key signals of **3** correspond to C12 at *δ*61.50 ppm and C13 at *δ*49.00 ppm, changes (comparing the C sp^2^-double bond) which were promoted by the oxygen atom; a low frequency for C13 is promoted by the protection of both methyl groups. The chemical shifts for other carbon atoms do not show significant changes. The spectroscopical characterization data of all the target compounds are displayed in the Appendix A.

### 2.4. Cytotoxic Evaluation of the Target Molecules

To know if **3**–**10** have pharmacological activity in comparison to perezone [12,13], their cytotoxic evaluation was performed through an MTT assay, using MDA-MB-231 breast cancer cells and normal fibroblasts as controls. It is important to note that the evaluation of the cytotoxic effect was carried out for all molecules, except for **8**, due to its poor solubility. Figure 2 shows the result for the most active compound (**6**) as a representative example. In addition, the results for compounds **3**–**5**, **7**, and **9**–**10** can be observed in the Appendix A.

In general, the perezone derivatives resulted in higher activity in comparison to isoperezone derivatives. However, the monoacetylated diol **6** shows the best pharmacological activity. In this sense, **6** displays an IC_50_ of 8.44 µM, showing a lesser mortality percentage against primary dermal fibroblast normal cells (18.39%). The target molecule **3** establishes a mortality of 19.51% with an IC_50_ of 9.62 µM. In the cases of **4** and **9**, their corresponding IC_50_ increased, and mortality diminished, with values of 114.39 and 100.53 µM in addition to 42.38 and 46.41%, respectively. The rest of the compounds show a higher mortality percentage. It is relevant to mention that the mean inhibitory concentration results described above are summarized in Table 1.

### 2.5. In Silico Study of Molecules ***3***–***10***

#### 2.5.1. Determination of Geometric and Spectroscopic Parameters

The optimization of the target molecules and the most stable conformer was supported using Spartan 06 [14]. The selected geometries were appropriately optimized in Gaussian 09 [15] using the DFT level, with the functional B3LYP and the basis set 6-311++G(d,p). The most stable conformer obtained in the gas phase for **3** and **10**, as representative examples, is shown in Figure 3. Additionally, the most stable conformers for **4**–**9** can be consulted in the Appendix A. In this sense, the transformation of the double bond of the side chain results in the generation of a chiral center in the carbon labelled C12; see Figure 1. Consequently, in this section, the optimization of **3**–**10** was performed considering both stereoisomers *(R* and *S)*.

The optimized structures, in general, show an intermolecular hydrogen bond between the hydroxyl in position C3 or C6 and the adjacent carbonyl group. Figure 3 displays the hydrogen bond for **3** and **10**; a better example is obtained in Figure 4 for **7** and **9** as representative examples. Here, it is important to highlight that **7** displays two hydrogen bonds; the other hydrogen bond was established between the hydroxyl group at position C12 and the carbonyl group of the ester in both stereoisomers, appearing with greater strength in the *R* stereoisomer; for the *R* isomer of **9** three hydrogen bonds can be observed, in which the second hydrogen bond was formed between OH3 and OH12, and the third hydrogen bond between OH12 and OH13.

Thus, according to Soriano-García et al. [16], the hydrogen bond generated between hydroxyl and carbonyl groups is responsible for the crystalline structure of perezone; consequently, the formation of other hydrogen bond interactions and the modification of the double bond probably produce solid amorphous forms for **3**–**6**, **8**, and **10** and oils for **7** and **9**. Additionally, **3**–**6** show cyclic systems of three and five atoms, which avoid rotation around side bonds, facilitating solid-state packaging.

The geometric parameter bond length of **3**–**10** (*R* and *S*) was calculated from the optimized structures (Table 2 and Table 3). It is important to highlight that the theoretical values obtained for this work were compared with experimental single-crystal X-ray data [16,17].

As can be noted in Table 3, the bond lengths for derivatives of perezone do not show significant changes in the quinone ring. But it is important to note that the hydrogen bond, a stronger bond, between the hydroxyl group OH3 and the oxygen C4=O of the carbonyl group decreases its distance (2 Å) [16].

Related to the modification of the double bond of the side chain, the distance displays an important increase, probably due to a change in the hybridization. For **3**, the theoretical values were 1.480 Å and 1.477 Å for the *R* and *S* isomers, respectively. Sankar et al. [18] reported for the epoxide from 5-fluoro-2-nitrobenzaldehyde an experimental bond length of 1.474 Å. For C−O bonds, **3** shows values of bond lengths of 1.439 Å for (*R*) C12−O, 1.448 Å for (*R*) C13−O, 1.442 Å for (*S*) C12−O, and 1.446 Å for (*S*) C13−O. Sankar et al. [18] reported experimental distances for C12−O of 1.439 Å and for C13−O of 1.435 Å.

For **5**, the observed bond lengths for C12−O and C13−O of 1.431 Å and 1.446 Å for *R* and 1.430 and 1.447 Å for *S* were compared with the ketal pentaerythritol of cyclohexanone [19], showing values of 1.430 and 1.425 Å. Additionally, the oxygen–carbon–oxygen bond distance of the acetonide moiety displays values of 1.428 Å and 1.438 Å for *R* and 1.429 Å and 1.438 Å for *S*, which were compared with the pyranoses [20], showing experimental values of 1.417 and 1.409 Å.

For **7**, the C12−O shows distances of 1.417 Å and 1.419 Å for the *R* and *S* stereoisomers, respectively, and for C13−O distances of 1.500 Å and 1.491 Å for the *R* and *S* stereoisomers, respectively. These distances were compared with experimental values [20]; thus, for C12−O, 1.432 Å was reported, and for C13−O, 1.477 Å was reported. For the C16−O13 bond of the ester moiety, distances of 1.336 Å and 1.337 Å were observed for the *R* and *S* stereoisomers, respectively, and the experimental value was 1.336 Å. For the C16=O of the ester moiety, the theoretical value for *R* and *S* stereoisomers was 1.215 Å, and the experimental value was 1.196 Å [20]. For the C16−C17 bond, the theoretical values were 1.509 Å for *R* and 1.508 Å for *S*, and the experimental literature value was 1.497 Å [20].

For **9**, the theoretical values for the *R* and *S* stereoisomers were 1.438 Å and 1.443 Å for C12−O, and for C13−O they were 1.435 Å and 1.437 Å. Ermer et al. [21] reported experimental values for C−O of 1.435 Å and 1.427 Å.

On the other hand, Table 3 summarizes the theoretical bond length values for derivatives of isoperezone; the quinonic ring displays similar values to those of isoperezone [17]. But it is important to note that intramolecular hydrogen bonds of **4**, **6**, **8**, and **10** show lower values in comparison to the experimental values reported for **2**; this is probably due to higher steric hindrance because the methyl group of the side chain is close to the carbonyl group, diminishing the distance between this group and the hydroxyl group.

Regarding the modifications performed on the double bond of the side chain, they were evaluated by contrasting them with the data established for the derivatives of perezone [18,19,20,21], concluding that in all cases the bond lengths present similar values.

#### 2.5.2. Theoretical–Experimental Correlation of ^1^H and ^13^C Nuclear Magnetic Resonance Chemical Shifts

The experimental values corresponding to the ^1^H and ^13^C NMR chemical shifts in the gas phase were correlated with the values calculated by DFT with B3LYP, as well as using the basis set 6-311++G (d,p) and the GIAO method. All data for experimental and theoretical results are in the Appendix A.

The chemical shift results were compared with experimental data previously obtained to determine if the level of theory applied in the present work is adequate for the determination of spectroscopic parameters of the molecules under study. A methodology used in the literature [22,23] determines the level of approximation between the theoretical and experimental results, involving a linear regression between the calculated and experimental parameters; thus, based on the regression coefficient value, the level of fit present in the data is obtained. The ^1^H NMR chemical shifts calculated by GIAO were plotted against the experimental data obtained for the perezone and isoperezone derivatives (Figure 5a,b).

**Figure 5 molecules-30-04603-f005:**
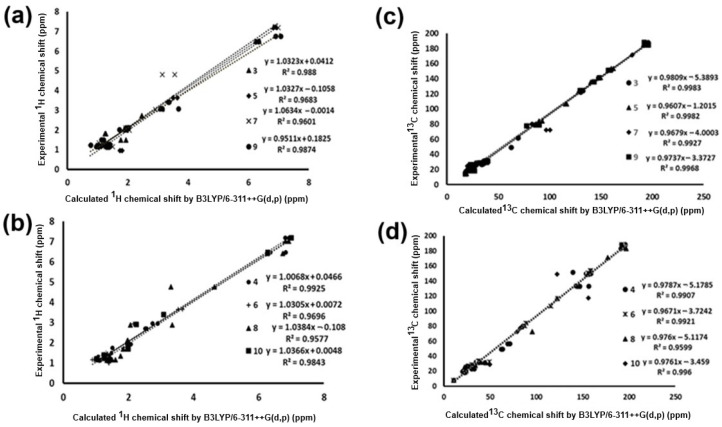
Linear regression between experimental and calculated values with B3LYP/6-311++G(d,p): ^1^H NMR chemical shifts for (**a**) perezone derivatives **3**, **5**, **7**, and **9** and (**b**) isoperezone derivatives **4**, **6**, **8**, and **10**. ^13^C NMR chemical shifts for (**c**) perezone derivatives **3**, **5**, **7**, and **9** and (**d**) isoperezone derivatives **4**, **6**, **8** and **10**. The atypical values for hydrogen of the hydroxyl group for compounds **7**–**10** were removed to obtain a better linear correlation; see Table 4.

**Table 4 molecules-30-04603-t004:** Regression equations and coefficients for (**a**) ^1^H and (**b**) ^13^C chemical shifts for **3**–**10**.

Molecules	Equations	Quantity of Hydrogen Removed from the Hydroxyl Groups	Regression Coefficient
(**a**)			
**3**	*δ*_experimental_ = 1.0323*δ*_Theoretical_ + 0.0412	0	0.9880
**4**	*δ*_experimental_ = 1.0068*δ*_Theoretical_ + 0.0466	0	0.9925
**5**	*δ*_experimental_ = 1.0327*δ*_Theoretical_ − 0.1058	0	0.9683
**6**	*δ*_experimental_ = 1.0325*δ*_Theoretical_ − 0.0072	0	0.9696
**7**	*δ*_experimental_ = 1.0634*δ*_Theoretical_ − 0.0014	1	0.9601
**8**	*δ*_experimental_ = 1.0384*δ*_Theoretical_ − 0.1080	1	0.9577
**9**	*δ*_experimental_ = 0.9511*δ*_Theoretical_ + 0.1825	2	0.9874
**10**	*δ*_experimental_ = 1.0366*δ*_Theoretical_ + 0.0048	2	0.9843
(**b**)			
**3**	*δ*_experimental_ = 0.9809*δ*_Theoretical_ − 5.3893	-	0.9983
**4**	*δ*_experimental_ = 0.9787*δ*_Theoretical_ − 5.1785	-	0.9907
**5**	*δ*_experimental_ = 0.9607*δ*_Theoretical_ − 1.2015	-	0.9982
**6**	*δ*_experimental_ = 0.9671*δ*_Theoretical_ − 3.7242	-	0.9921
**7**	*δ*_experimental_ = 0.9679*δ*_Theoretical_ − 4.0003	-	0.9927
**8**	*δ*_experimental_ = 0.9760*δ*_Theoretical_ − 5.1174	-	0.9599
**9**	*δ*_experimental_ = 0.9737*δ*_Theoretical_ − 3.3727	-	0.9968
**10**	*δ*_experimental_ = 0.9761*δ*_Theoretical_ − 3.4590	-	0.9960

The graph allows us to observe that at this level of theory and under the applied models, molecules **3**, **4**, **9**, and **10** present a better fit. In the case of the hydroxyl groups, the points are far from linear, and they were removed from the graphs because the theoretical values are atypical, consequently diminishing the regression coefficient value. This is because, in the experimental spectrum, the hydroxyl groups exhibit a common dissociation, a fact that does not occur in the gas phase spectrum [24].

The linear regression analysis for the ^1^H NMR data (Figure 5) provided the linear equations for these correlations, which describe the behavior of perezone and isoperezone derivatives at the level of theory evaluated (Table 4a). The results show values of regression coefficients greater than 0.9500, indicating that the methodology used allows adequate description of the experimental data.

The same analysis was performed for the theoretical and experimental chemical shift values for ^13^C NMR; the corresponding graph is shown in Figure 5c,d. All data for experimental and theoretical results are in the Appendix A. The linear regression analysis for the ^13^C chemical shifts gave regression coefficients in orders greater than 0.960, and the equations are summarized in Table 4b.

Considering these graphs, both analyses reflect a good description between theoretical and experimental ^1^H and ^13^C NMR chemical shifts by the selected method, theory level, and basis set (Table 4).

#### 2.5.3. Molecular Orbital Analysis

In the literature, a relationship has been established between structures with electrophilic groups and structures with nucleophilic groups to explain the biological activity of proteins, nucleic acids, and other metabolites. In addition, around 266 known drugs are generated from natural molecules that have electrophilic sites such as Michael-type acceptors, stressed structures, esters, and carbamates, among others [25]. In this sense, the theoretical study of the compounds, employing their molecular orbitals, was focused on the behavior of the LUMO orbitals and their stability through the GAP (or ELUMO-EHOMO) of the molecular orbitals in order to consider the possible sites of interaction, allowing for explanation of a pharmacological effect.

Figure 6a shows the energy difference values between the calculated border orbitals (GAP), and Figure 6b displays the energy levels of the LUMO orbitals. The comparison between GAP values reflects a constant tendency in which isoperezone derivatives were less reactive than perezone analogs [12,26]. In this sense, it can be appreciated that *S* stereoisomers are less reactive than *R* for both perezone and isoperezone derivatives. Thus, to determine the electrophilic character of the molecules studied, the energetic values of their LUMO orbitals were compared.

Figure 6a exhibits that **4**-*S* displays less reactivity, with a ΔE_GAP_ = 79.40 kcal/mol, being the most stable compound, followed by **8**-*R* (ΔE_GAP_ = 79.30 kcal/mol) and **10**-*S* and **4***-R* (ΔE_GAP_ = 79.20 kcal/mol). The high stability of **8** and **10** was attributed to the presence of less reactive substituents, hydroxy or monoester groups, at C12 and C13. On the other hand, the most reactive molecules were **9**-*R* (ΔE_GAP_ = 73.40 kcal/mol) and **5**-*R* (ΔE_GAP_ = 73.50 kcal/mol).

Correlating these results with the IC_50_ values (Table 1), it can be observed that the reactivity of **6** is slightly higher than that of **4** and **10**; consequently, the IC_50_ of **6** (8.44 μM) against breast cancer cells was higher than those of **4** and **10** (114.39 and 139.75 μM, respectively). For **5**, a lower E_GAP_ value was presented, with these being the most reactive molecules, but with intermediate pharmacological activity among all the compounds, with an IC_50_ value of 74.91 μM. These facts could probably be attributable to different interactions between substituents of derivatives and residues of amino acids of the proteins of cancer cells.

Regarding LUMO orbitals (Figure 6b), they are a parameter that describes the reactivity and electrophilicity of the molecules; thus, a low value of ELUMO is related to the ability of the molecule to accept electrons. The electron density surfaces for **3**-*S* and **6**-*R*, as representative examples, were obtained for the LUMO molecular orbitals and are shown in Figure 7. The LUMO orbitals for all molecules were located around the quinone ring; they can be viewed in the Appendix A. Higher electrophilicity (Figure 6b) was observed for **9**-*R*, **9***S*, and **3**-*R*; therefore, they could be more susceptible to an attack by a nucleophile compared to the other study systems. Further, the structures that showed less electrophilicity were **8**-*R* and **4**-*R*. The foregoing is consistent with the cytotoxicity data, with **6** and **3** being the most active molecules.

#### 2.5.4. Molecular Electrostatic Potential Maps

The electrostatic potential map is a property used to appropriately predict molecular reactivity as well as carry out studies of the interaction between compounds of interest and molecules of biological origin [27]. It has also been considered as an indicator of the reactivity regions of a target molecule and has been used to study the interactions between electron donors and electron acceptors—for example, a drug and the amino acid residues of the cellular receptor [27,28].

Consequently, this property was calculated for all molecules with their corresponding stereoisomers using the B3LYP/6-311++G(d,p) level of theory, with a value range from ±6.0 × 10^−2^ to ±4.2 × 10^−2^ for electrophilic (blue color) and nucleophilic (red color) sites. The corresponding maps are shown in Figure 8 for **3**-*S* and **6**-*R*; the other maps are in the Appendix A. The distributions indicate that in all cases, the oxygen atoms of the carbonyl groups of the quinone ring are in a region with higher electron density, as are the oxygen atoms of the substituents between the C12−C13 bond, but it is important to highlight that the oxygens of the hydroxyl groups, positions C6, C12, and C13, have a lower distribution due to the intramolecular hydrogen bonds.

On the other hand, most of the electronic deficiencies for all molecules are placed on the hydrogen atom of the hydroxyl group of the quinone ring and the hydroxyl groups generated at the C12−C13 bond. Consequently, it is important to highlight that these atoms could participate in non-covalent interactions with some residues of the receptors.

#### 2.5.5. Prediction of Toxicological and Physicochemical Properties

Perezone and isoperezone are molecules of synthetic interest due to their pharmacological properties and their possible use for the generation of biologically active molecules; moreover, perezone exhibits coupling with caspases, molecules responsible for programmed cell death.

Chemo-computer studies were carried out for all compounds, employing the Osiris property explorer [29] (Table 5). It is important to note that this database does not distinguish between stereoisomers; consequently, the results are for both stereoisomers. The toxicological risk results indicated that quinones **1** and **2** display high irritation activity, and compounds **7** and **8** show low irritation activity. Also, it is important to note that compounds **3** and **4** display high reproductive effects and low tumorigenic activity. In consequence, the effects could be attributed to the presence of the epoxy moiety in the C12−C13 bond.

The adverse effects predicted for **3** and **4** are consistent with some reports presented in the literature, indicating that epoxides can cause DNA damage and generate tumors [30] or damage the genetic material of gametes [31], and consequently cause reproductive problems.

Concerning physicochemical properties, for *cLogP* (<5) [28], the logarithm of the partition coefficient between *n*-octanol and water, a property that describes the hydrophobic character of the molecule, **1** and **2** present higher values (3.31) than **5** and **6** (2.49) and **3**, **4**, and **7**–**10** (1.88, 1.78, and 1.29, respectively) [32]. The values are less than 5, which indicates the probability of being well-absorbed [29]. The solubility of the drug (expressed as Log S) also allows the description of the absorption process. Poor solubility leads to poor absorption and bioavailability [27,32,33,34]. Commercial drugs have *LogS* values greater than −4. In this sense, all compounds show values within that range, from −2.97 to −2.00, indicating that these compounds can have adequate absorption, movement in the bloodstream, and elimination through the urinary tract. Compounds **5** and **6** present the lowest solubility (−2.97), which is consistent with the *cLogP* values that confirm a lower affinity to water compared with **3**, **4**, and **7**–**10**.

Regarding molecular weight, all compounds have values greater than 160 but less than 500 Da, a result that is suitable for a potential drug according to Lipinski’s rule of five [34]. In addition, another good descriptor of absorption is total polar surface area (TPSA), which includes intestinal absorption, bioavailability, Caco-2 permeability, and penetration of the blood–brain barrier [35]. These molecules present values that can be comparable with those of other drugs. The TPSA value for all cases was less than 140, indicating that all the molecules show adequate values for penetration through the cell membrane. Additionally, it can be observed that only raw materials have adequate values for penetration of the blood–brain barrier (<60 Å^2^) [36].

The drug score (DS) is the combination of the similarity of the studied molecules with drug-likeness, *cLogP, LogS*, molecular weight, and risks of toxicity in a practical value that could be used to judge the compound’s potential to qualify as a drug [37]. In this regard, **10** presented the best drug score, followed by **6**, which, as mentioned, is one of the most biologically active compounds against cancer cells, while **3** and **4** presented low values. The lower drug score predicted for **3** and **4** was the consequence of the high probability of reproductive effects and the low probability of tumorigenic effects in these molecules [36,38].

Finally, regarding acceptors as well as donors in hydrogen bonding interactions, Lipinski’s rule of five indicates that there must be five donors and less than ten acceptors to avoid bioavailability problems; this fact is fulfilled for all molecules. Consequently, all compounds must be considered to perform in vivo studies [30,39].

The results also confirm that modifications in the double bond of the side chain improve the properties of the molecule, as was established in previous sections. In a complementary way, the shape index, molecular flexibility, and molecular complexity were determined. The first parameter indicates that the molecules studied present a linear geometry (>0.5), while all molecules present high flexibility and molecular complexity (>0.5).

#### 2.5.6. Study of Molecular Coupling in the Apoptosis Pathway

It is relevant to highlight that in previous research, it was shown that **1** and **2** and other sulfur derivatives activate the apoptosis pathway [12]. Thus, the molecules in this study probably activate a possible apoptosis pathway. In this sense, a study of the docking of the compounds studied with proteins participating in the apoptosis pathway was performed [40]. Additionally, considering that the effector molecules of apoptosis include caspase-3, in the present study, the interaction between the synthesized molecules and caspase-3 was evaluated to explain its cytotoxic effect and their differences in activity.

The results of the molecular coupling obtained from Autodock vina 4.2 [41] and visualized in Discovery Studio Visualizer 2019 [42] for the target compounds with caspase-3 are shown in Table 6, where the binding energy between the protein and the target molecule is presented, considering each of its stereoisomers.

In the first instance, it can be pointed out that, according to the values obtained, the new derivatives presented a higher interaction energy than **1** and **2**. The molecules with the highest energy were **6**-*R* (−8.14 kcal/mol), **6**-*S* (−8.04 kcal/mol), and **5**-*R* (−8.00 kcal/mol), while the molecules that showed the least interaction with caspase-3 were **4** (−7.06 and −7.10 kcal/mol for *S* and *R*, respectively), **9** (−6.72 and −7.02 kcal/mol for *R* and *S*, respectively), and **10** (−6.65 and −6.75 kcal/mol for *S* and *R*, respectively).

Bearing in mind that **6** presented experimentally the highest biological activity, followed by **3**, the above is of great relevance since the theoretical results are consistent with the experimental results, in which **6** (−8.14, −8.04 kcal/mol) showed the most stable binding energy, and **5** (−8.00, −7.82) and **3** the next best affinity (−7.29, −7.22), allowing explanation of the differences in cytotoxicity. In the case of **6**-*R*, the proposed molecular coupling models are presented in Figure 9a,b. As the mode of attachment of the molecule with the protein is through interactions with both catalytic site residues, the first ones between one of the quinonic ring oxygens and the adjacent hydroxyl group, with Cys163, showed a 2.87 Å hydrogen bond and a π-donor hydrogen bond between the quinonic ring and the Cys163 sulfhydryl group. The third interaction occurs in the same way as a double interaction between the mentioned oxygen atom and the quinonic ring with His121, including a π-π T shape interaction in the quinonic ring with the imidazole ring from the same residue. These interactions are shown because, in the electrostatic potential maps, it is observed that the quinonic ring oxygens are areas of high potential, allowing hydrogen bond interactions, in addition to the quinonic system, which presents an electronic excess, which would allow π-π and π-donor interactions.

For the **3**-*R* epoxide (Figure 9c,d), it was observed that the interaction of this compound with the protein is also governed by one hydrogen bridge-type interaction, established between the oxygen from the carbonyl group at C1 and the imidazole ring from His121 at 2.94 Å. This interaction can be classified as a moderate hydrogen bond. The second most relevant interaction is established as a π-donor hydrogen bond between the perezone quinone ring and sulfhydryl group from Cys163. Interactions with both catalytic site residues are presented as well; however, for this compound, a greater number of additional interactions all over the catalytic site on behalf of the epoxide modification are shown, mainly being π-alkyl and π-sigma interactions, including a lone pair-π interaction between the Trp206 ring and the epoxide oxygen. The compound’s affinity to the catalytic site can support the observed biological effect.

For **5** (Figure 9e,f), three main interactions were generated between the protein and the ligand, the first one between the carbonyl group of C1 and the cysteine residue 163, with this residue being considered as the hydrogen donor whose distance is established at 2.68 Å. Therefore, it is shown that the interaction occurs through a moderate hydrogen bond, given its distance. The second and third interactions are established at the quinone ring, where a π-donor hydrogen bond between the Cys163 sulfhydryl group and the mentioned ring was formed, in addition to a π-π T shape interaction with the imidazole ring from His121, the first residue from the catalytic site. Cys163, the second catalytic site residue, resulted in a double interaction with this compound.

Regarding biological activity, the next compound in order of activity was **9** with an IC_50_ of 100.53 μM; in this respect, the theoretical study showed a decrease in binding affinity, since both stereoisomers presented binding energies of −7.26 and −7.04 kcal/mol, explaining the observed increase in IC_50_ experiments, leading to a higher compound concentration to achieve the same effect.

The interaction model (Figure 9g,h) for **9**-*S* is proposed through the formation of four hydrogen bonds established in the C12™C13 diol moiety, and one more in the quinone ring. From these interactions, only one hydrogen bond corresponded to a residue from the catalytic site (His121). It is worth noting that the diol region modification allowed several more different interactions with amino acid residues around the catalytic site; however, the mentioned interactions did not improve the binding affinity.

In the case of the diol moiety, the hydrogen bridge is found with the His121 residue from the catalytic site, exhibiting a 3.17 Å distance, where the imidazole ring acts as a donor, while the hydroxyl group at the diol derivative acts as an acceptor. The hydrogen bond is considered a weak interaction. At the quinone ring, the interaction with Cys163 is reported as a π-sulfur interaction, involving the sulfhydryl group and the pi electrons from the quinone ring. It is important to note that the perezone monoacetylated diol **8** presented similar (but slightly higher) values in interaction energy than those presented by **9**; however, in order of cytotoxicity, it was one of the less active (33 µg/mL), which can be attributed, according to the properties calculated in OSIRIS Data Warrior, to lower solubility and a higher TPSA value, which reduces bioavailability as well as penetration through the membrane.

For **8**-*R*, the proposed binding with caspase-3 is shown in Figure 9i,j. The results confirm that the first relevant interaction is located between the imidazole ring from His121, which acts as a hydrogen donor, and the hydroxyl group at the quinone ring, which acts as an acceptor. The mentioned interaction is a hydrogen bond, which has established a weak interaction with 3.26 Å. Finally, the last important interaction is related to Cys163, where a hydrophobic π-alkyl interaction was found with the quinone ring. It is worth mentioning that isoperezone monoacetylated diol **7** resulted in similar binding energy values to **8** (−7.11 and −7.27 kcal/mol, respectively), even establishing hydrogen bridges and hydrophobic interactions with one or both catalytic site residues from caspase-3. However, IC_50_ experiments demonstrated that **7** did not show great solubility conditions in testing, which means it is not desirable as a good drug candidate.

Moreover, isoperezone epoxide (**4**) presented lower biological activity than perezone diol **9** (30 µg/mL), but slightly higher activity than perezone monoacetylated diol **7**, having resulted in binding energies of −7.10 and −7.06 kcal/mol. The principal observed interactions at the catalytic site are found between His121 as a π-π T-shaped interaction and a π-donor hydrogen bond with Cys163 (Figure 9k,l). It is remarkable to mention the observed binding differences among perezone and isoperezone epoxides, where hydroxyl group position changes in the quinone ring considerably decreased the binding energy and interaction type within the protein pocket. The modification in the C12™C13 bond did not contribute to better coupling or affinity for this compound towards caspase-3.

To finalize the coupling study, the least active molecule turned out to be the isoperezone diol **10**, which presented interaction energy values of −6.75 and −6.65 kcal/mol, which are consistent with the theoretical determinations, in which this molecule is marked with the highest GAP, which is the most stable. An additional feature to note is that it has a lower interaction energy than **4**, explaining its cytotoxic activity decrease (39.41 μg/mL) as well, with it being the least active compound of all the studied derivatives.

The proposed interaction model (Figure 9m,n) consists of a π-π T-shaped interaction as well, with His121 and the recurrent π-donor hydrogen bond between Cys163 and the quinone ring in the isoperezone derivative **10**. The presence of these interactions did not allow for achieving a great binding energy, which happened in general when there was not at least one hydrogen bond interaction with one of the catalytic site residues; this molecular docking result was also supported by and concluded with the lowest IC_50_ from all proposed derivatives.

Also, it is important to mention that some previous manuscripts reported binding energy values, employing *N*-acetylcysteine as a reference, from −0.6 to −9.4 kcal/mol [43], and assessed triazoles for their anti-apoptotic activity in tests. Another interesting study reported that the complex plumbagin-caspase-3 displays a binding energy of −10.13 kcal/mol, assessed in lung cancer cells [44]. Additionally, the binding energy between D-galacturonic acid and caspase-3 was −4.03 kcal/mol [45], assessed for diabetes mellitus. Considering these results, the binding energy of molecules **3**–**10** obtained in this work displays similar energies, offering good cytotoxic activity, as seen above.

#### 2.5.7. Prediction of Pharmacological and Metabolic Properties

An additional chemo-computer study for all compounds to predict their absorption, metabolism, and excretion properties in humans was carried out, considering the potential use of the molecules studied as drugs. In this sense, the behavior in different absorption and excretion models was studied using the ADMETSAR methodology [46], summarizing the results in Table 7 for **1** and its derivatives; the data for **2** and its derivatives can be found in the Appendix A. The proposed predictions for all derivatives reveal human intestinal absorption with probability values greater than 0.700, presenting better probability values for isoperezone derivatives. Regarding the blood–brain barrier, all compounds show absorption probability values greater than 0.700, with the molecules with the highest probability being **3**, **5**, and **7** (0.913, 0.903, and 0.804, respectively), and **9** had the lowest probability.

With regard to P-glycoproteins, proteins involved in the elimination of xenotoxins against pronounced concentration gradients, as indicated above, all derivatives showed a good probability as substrates, presenting the highest probability for **9** and **10**. Moreover, this property is relevant already in that P-glycoproteins play an important role in the transport of small molecules in vital areas and are present in cancer cells with resistance to multiple drugs, their inhibition being crucial to overcoming this type of resistance [46].

The behavior of **3**–**10** as substrates of P-glycoprotein is related to the fact that their structures do not fulfill the “Rule of Four” [34], due to having masses less than 400 Da and containing at least four oxygen atoms in their structure. The prediction of the inhibitory character of **3**–**10** on P-glycoprotein I and II indicated that **4**, **6**, and **8** are inhibitors of glycoprotein I, while **3**–**8** only act as inhibitors of glycoprotein II. In the case of **9** and **10**, they did not show an inhibitory character in the P-glycoproteins. Finally, none of the molecules showed an inhibitory effect on the renal organic cation transporter.

Other interesting results included the prediction of the metabolism of **1** and its derivatives (Table 8) using the methodology of Admetsat and Metaprint2D; the results for **2** and its derivatives can be found in the Appendix A. In the first place, the behavior as a substrate or inhibitor of the molecules synthesized in the most important isoforms of the cytochromes P450 was evaluated. During the biotransformation of drugs, the molecules break down and/or convert into more soluble molecules, which play an important role in the pharmacokinetic and therapeutic action of drugs.

Evaluation of the target molecules as CYP substrates shows that **3**–**10** are not substrates of the 2C9 and 2D6 isoforms of cytochrome P450. The results obtained for these cytochromes can be explained by considering the structure that their substrates generally present; in the case of CYP2C9 substrates, they have weak acidic properties and multiple aromatic rings [47]. All molecules have a weakly acidic hydroxyl group; however, they do not have aromatic rings. Considering CYP 2D6, the main characteristic of its substrates is the presence of a basic nitrogen atom placed at 5 Å or 7 Å from the oxidation site [48], which would exclude the molecules in this work because they do not have nitrogen atoms.

The prediction of the behavior of the substrate in CYP3A4 indicated that all the molecules could be substrates because these molecules fulfill the pharmacophore model features for this cytochrome. Molecules **9** and **10** present, considering MEP and charge data, two hydrogen bonding acceptors (C1=O and C4=O), one hydrogen bonding donor (C3-OH, C6-OH, OH12, or OH13), and a hydrophobic region (side chain from C8 to C15). Additionally, an important metabolic property to consider is the ability of these compounds to inhibit CYP isoforms, which is an adverse side effect. In this sense, non-inhibitory behavior is predicted by the target molecules.

#### 2.5.8. Metabolism of Human Phase I Perezone and Isoperezone Derivatives

To conclude the present research, a study of possible sites conducive to phase I metabolism (basically defined as hydroxylation, oxidation, epoxidation, or elimination reactions) of the target molecules was performed, using metaprint2D in the application Bioclipse, which is a tool that is based on historical data on the metabolism of various molecules [49,50]. The results are shown in Figure 10.

The colors highlight atoms, indicating their normalized rate of occurrence (NOR). The red color indicates high NOR values (from 0.66 to 1), the orange color designates values between 0.33 and 0.66, the green color indicates a range of NOR values from 0.15 to 0.33, the white color indicates values from 0.00 to 0.15, and gray denotes missing data. For **3**–**8**, they recurrently showed C14 and C15 as metabolic centers, with NORs ranging from 0.15 to 0.66. Here, it may be proposed that these carbons undergo hydroxylation and even their transformation into carboxyl groups [34,50]; the same behavior could be proposed for C16 and C17 for **5** and **6**.

Related to the isoperezone derivatives, the reactivity in C2 allows for supposing there is an epoxidation in the double bond of the quinone; this site generally has a high NOR, but epoxidation is not possible when the hydroxyl is in the C3 position, as is the case for **1**, in which this reactive site is not present. Additionally, **3**–**8** show in C7 moderate to low NOR (orange to green); due to this, carbon is highly susceptible to being hydroxylated by radical species. It was also observed that C12 has a high NOR at **7**–**10** because the carbon supports a hydroxyl group that can be oxidized to a carbonyl compound.

Finally, it is important to point out that diols **9** and **10** were the molecules with the fewest metabolic sites in phase I, because they are the molecules with the best-estimated solubility in water; thus, their excretion would be easier in comparison with the rest of the compounds. In this sense, the targets **5**–**8** present several sites for metabolism, consistent with their solubility values.

## 3. Materials and Methods

### 3.1. Materials and Equipment

Starting materials: Urea hydroperoxide and immobilized lipase B *Candida antarctica* (Novozyme 435) were purchased from Sigma-Aldrich Chemistry (St. Louis, MO, USA) and used as received. The anhydrous solvents, ethyl acetate and acetone, were prepared according to the method in the literature [51]. The solvents *n*-hexane and ethyl acetate were technical grade and were used without further purification, and were from Materiales y Abastos Especializados S.A. de C.V. (Zapopan, Jalisco, Mexico). Perezone was isolated from *Acourtia platyphilla* specimens and identified by its corresponding spectroscopical data correlated with the literature [2]. Isoperezone was produced according to the procedure in the literature [52]. The reactions were monitored by employing thin-layer chromatography (TLC) performed on pre-coated Merck silica gel 60F254 aluminum sheets, and visualization was achieved using a 254 nm UV lamp (UVLS-24, Upland, CA, USA), and purification was achieved with column chromatography using silica gel (0.0063–0.200 mm, 70–230 mesh ASTM, acquired from Merck-Millipore, Darmstadt, Germany). ^1^H and ^13^C NMR spectra were recorded using a Varian Mercury-300 spectrometer (Palo Alto, CA, USA) at 300 MHz and 75 MHz for hydrogen and carbon, respectively. The multiplicities were reported as singlet signal (s), singlet broad signal (bs), doublet signal (d), triplet signal (t), and multiple signal (m). The corresponding *δ*-chemical shifts are given in ppm, using CDCl_3_ as a solvent, and TMS as an internal reference. The EI-MS (70 eV) and HRMS-EI were determined using a JEOL JMS-700 MStation mass spectrometer (JEOL, Tokyo, Japan). Elemental composition was calculated within a mass range of ±10 ppm from the exact measured mass. Melting points were determined using a Fisher-Johns apparatus (Fisher Scientific, Waltham, MA, USA) and are uncorrected. Microwave-assisted production of target compounds was performed using CEM Focused Microwave^®^ Synthesis System microwave equipment (Matthews, NC, USA). The IC_50_ statistical analyses were performed with the PRISMA statistical program [53]. The theoretical calculations at the quantum level were carried out using the GAUSSIAN 09 program (Version 09, Gaussian, Inc., Wallingford, CT, UK, 2013) [54], while the molecular coupling studies were developed using Autodock 4.2 [41].

### 3.2. Molecules Obtained by Oxidation of the C12™C13 Bond

#### 3.2.1. Production of Epoxides **3** and **4** from Perezone (**1**) and Isoperezone (**2**)

The corresponding spectroscopic characterization and comparative signals between theoretical and experimental chemical shifts can be consulted in the Appendix A.

In an appropriate glass vessel, 1.008 mmol (250 mg) of perezone or 1.008 mmol (250 mg) of isoperezone was mixed with 2.016 mmol of 97% urea hydroperoxide (190 mg), Novozyme^®^ 435 (30 mg), and 5 mL of anhydrous ethyl acetate, and then the mixtures were stirred at room temperature for 5 h. The reaction progress was monitored using TLC, with a mixture of *n*-hexane/ethyl acetate (90:10) as an eluent. After this, the mixture was washed with anhydrous ethyl acetate, and the solvent was partially evaporated under reduced pressure. The product was purified by column chromatography using silica gel as a support and the system *n*-hexane/ethyl acetate (80:20) as a mobile phase.

#### 3.2.2. Acetonides **5** and **6** from Epoxides **3** and **4**

In an appropriate glass vessel, 1.023 mmol (270 mg) of **3** or 1.023 mmol (270 mg) of **4** was mixed with 1250 mg of activated TAFF as the catalyst and 5 mL of anhydrous acetone, and the mixtures were stirred at room temperature for 19 h. The reaction progress was monitored using TLC, using as an eluent a mixture of *n*-hexane/ethyl acetate (70:30). After this, the mixture was filtered to remove the TAFF catalyst and washed with anhydrous ethyl acetate, and the solvent was partially evaporated under reduced pressure. The product was purified by column chromatography using silica gel as a support and, as a mobile phase, the system *n*-hexane/ethyl acetate (80:20).

#### 3.2.3. Monoacetylated Diols **7** and **8** from Epoxides **3** and **4**

In an appropriate glass vessel, 1.023 mmol (270 mg) of **3** or 1.023 mmol (270 mg) of **4** was mixed with 1250 mg of activated TAFF as the catalyst and 5 mL of anhydrous ethyl acetate. The mixtures were stirred at room temperature for 19 h. The reaction progress was monitored by TLC, using as an eluent a mixture of *n*-hexane/ethyl acetate (70:30). After this, the mixture was filtered to remove the TAFF catalyst, and the solvent was partially evaporated under reduced pressure. The product was purified by column chromatography using silica gel as a support and, as a mobile phase, the system *n*-hexane/ethyl acetate (80:20).

#### 3.2.4. Diols **9** and **10** from **1** and **2** or Epoxides **3** and **4**

##### Telescoped Process

In an appropriate glass vessel, 1.008 mmol (250 mg) of **1** or 1.008 mmol (250 mg) of **2** was mixed with 2.016 mmol of 97% urea hydroperoxide (190 mg), Novozyme^®^ 435 (30 mg), and 5 mL of anhydrous ethyl acetate**,** and then the mixtures were stirred at room temperature for 5 h. The reaction progress was monitored by TLC, with a mixture of *n*-hexane/ethyl acetate (90:10) as an eluent. Then, 3 mL of a 2:1 mixture (water/acetic acid) was added and stirred for 20 min. The reaction progress was monitored by TLC, with a mixture of *n*-hexane/ethyl acetate (90:10) as an eluent. After the reaction mixture was filtered and washed with anhydrous ethyl acetate, the solvent was partially evaporated under reduced pressure. The product was purified by column chromatography using silica gel as a support and the system *n*-hexane/ethyl acetate (70:30) as a mobile phase.

##### Hydrolysis Activated by Microwave

In a 50 mL round-bottom flask, 1.023 mmol (270 mg) of epoxides **3** or **4** was placed with 25 mL of water. The suspension obtained was irradiated with microwaves with a power of 250 W at 100 °C for 10 min. The resulting reaction mixture was extracted twice using 20 mL of ethyl acetate. The organic phase was dried over anhydrous sodium sulfate, and the solvent was evaporated at room temperature. The progress reaction was monitored by TLC using a mixture of *n*-hexane/ethyl acetate (70:30) as an eluent. The product was purified by column chromatography employing silica gel as a support and the system *n*-hexane/ethyl acetate (70:30) as a mobile phase.

### 3.3. Cytotoxicity Studies

#### 3.3.1. Cytotoxicity Test Using the 3-(4,5-Dimethylthiazol-2-yl)-2,5-diphenyl Tetrazolium Bromide Technique (MTT Test)

This test was performed using the MDA-MB231 breast cancer cell line (ATCC^®^ CRM-HBT-26), as described by Mosmann [55]. The cells were incubated at different concentrations of the compounds of interest. To generate the cell viability curves of all synthesized compounds, cells were seeded in 96-well culture plates at a density of 5000 cells/well and cultured in DMEM supplemented with 10% FBS. Cell viability was measured using MTT reagent dissolved in PBS (0.5 mg/mL). On the day of measurement, the medium was carefully replaced with fresh DMEM and 10% SFB containing diluted MTT (1:10, 10% MTT), and the sample was incubated for 1 h at 37 °C in a CO_2_ incubator to allow for the transformation of MTT dye into Formazan salt. After removing the incubation medium, the Formazan crystals were dissolved in 100 µL of DMSO solution. It is also important to note that the reduction in MTT was quantified by measuring absorbance at 570 nm using the Benchmark Plus absorbance microplate reader (Bio-Rad). The MTT test was repeated nine times. The proliferation percentage (% viability) was calculated according to the following:% Viability = (B/A) × 100
where A is the absorbance value of the control group and B is the absorbance value of the cells treated with the compounds of interest. The concentration of cisplatin, as a reference, was 40 µM, with a mortality of 40.53 ± 0.135% at 48 h of exposure, and primary dermal fibroblast; normal, human, neonatal (HDFn, ATCC^®^ PCC-201-0102) as a control at 24 h exposure.

#### 3.3.2. IC_50_ Determination

IC_50_ was extrapolated from the dose–response plot, obtained by plotting viability versus concentration. The concentration of the drug that reduced the viability of the cells by 50% (IC_50_) was determined by plotting the data points in triplicate over a concentration range and calculating the values using regression analysis.

### 3.4. Statistical Process Control

Analysis of variance was performed using the ANOVA procedure. Tukey’s test was used to determine the differences of means, and the significance of differences was evaluated at the *p* < 0.05 level. Data analysis was carried out using the GraphPad software PRISM statistical program 2020 (GraphPad by Dotmatics, Boston, MA, USA) [53].

### 3.5. Computational Theoretical Chemistry Studies

#### 3.5.1. Computational Theoretical Chemistry Studies at the Quantum Level

The reported calculations were carried out using Density Functional Theory (DFT) [56,57], using the exchange of three Becke parameters and the Lee–Yang–Parr correlation hybrid functional (B3LYP) [58,59] with the 6-311++G(d,p) basis set that includes divided valence functions and fuzzy functions [60,61]. The corresponding quantum mechanical calculations to determine the ^1^H and ^13^C chemical shifts were performed using the invariant atomic orbital estimation method (GIAO method) [62]. The energetic difference between the highly occupied molecular orbital and the least occupied molecular orbital (HOMO-LUMO), also known as GAP, is a typical quantity used to describe the dynamic stability of molecules [39]. The values of the energy of the orbital and the surface of the boundary orbitals were calculated using the same level of theory [28]. For the calculation of analysis of natural electronic populations, the methodology of the natural bonding orbital (NBO) present in the Gaussian 09 program [63] was used [64]. The charge analysis was carried out to explain the interaction models with the corresponding ligands [65]. Finally, molecular electrostatic potential (MEP) maps were calculated for the target molecules at the same level of theory to complete the electronic analysis, considering the importance of these results in the interaction models in the biological system studied [66]. All the figure results were visualized and analyzed using GaussView 5.0 [67].

#### 3.5.2. Molecular Coupling Simulation

Molecular docking simulations were performed using the three-dimensional crystal structure of caspase-3, retrieved from the Protein Data Bank (PDB ID: 6CKZ) [68]. AutoDock Tools 1.5.7 and Discovery Studio Visualizer 2019 [41] software were used to study all conformations and analyze the couplings. The parameters used for the study in AutoDock Tools were performed with an exhaustiveness value of 10, and the number of evaluated conformations was set to 100. The ligand was not rigid and was allowed to twist in the coupling processes. A grid-based approach was used to define the search space with a lattice size of 60 Å × 60 Å ×60 Å. The binding positions were grouped using the root of the mean standard deviation between the Cartesian coordinates of the ligand atoms and similar binding energy values. The binding modes with the most negative binding free energy were selected as the optimal coupling conformation. The binding results were presented graphically using Discovery Studio Visualizer 2019 [42]. The validation of the docking protocol was performed by re-docking the cocrystallized ligand (Ac-DW3-KE) from the original protein structure, obtaining an RMSD value of 0.5399 Å (Appendix A). Root-mean-square deviation was employed for difference evaluation of the obtained re-docking and cocrystallized pose of the same ligand molecule [69]. After docking validation, binding free energies from the studied compounds and *N*-acetylcysteine, usually used as a reference compound due to its known antagonist effect towards caspase-3 and oxidative stress, were calculated [70].

#### 3.5.3. Physicochemical Properties Studies

Predictions of physicochemical properties for the molecules under study provide valuable information on the ease with which a drug molecule acts on amino acid residues within cells or membrane receptors. The toxicological risk and the physicochemical properties of the studied molecules were obtained using the OSIRIS properties explorer as well as the OSIRIS-Data Warrior package. The toxicological risk prediction process is based on a recompiled set of structural fragments that give rise to toxicity alerts if they are found in the structure provided to the system. *LogP* and *LogS* were estimated using the OSIRIS method, which is implemented as an additive system of contributions for each atom based on its properties. The drug-likeness approach is based on a list of about 5300 different substructure fragments with associated drug-likeness scores. The similarity of the molecules to a drug is calculated using the score values of the fragments present in the molecule under investigation [26,29,35].

#### 3.5.4. Studies of Metabolism, Absorption, and Excretion

The absorption and metabolic properties of the studied compounds were calculated using the AdmetSAR server, which predicts about 50 ADMET endpoints (adsorption, metabolism, and excretion) using a chemoinformatic tool called the ADMET simulator, which integrates high-quality QSAR predictive models [46].

#### 3.5.5. Phase I and II Metabolism Studies

The human metabolism proposed for the synthesized compounds was proposed using metaprint2D-Reaction software, version 2.6.2, as well as the previous software module present in the Bioclipse package, which consists of a set of predictions of xenobiotic metabolism through data extraction and statistical analysis of transformations of known metabolic disorders reported in the scientific literature [71,72]. The results are shown in colored circular marks that predict the reactions at that site and the possible types of reaction, in some cases showing the metabolite formed. The color of the mark on the atoms indicates their normalized rate of occurrence (NOR) values. A high NOR value indicates a frequently reported site of metabolism in the metabolite database.

## 4. Conclusions

In this research on perezone, a sesquiterpene quinone recognized as the first natural product isolated in crystalline form on the American continent (1852), both the general and particular objectives originally established were accomplished; thus, in general, a green chemistry contribution to the chemistry of perezone was accomplished: C12−C13 bond oxidation, a theoretical study, and a cytotoxic evaluation of the obtained products were performed. In particular, this green contribution involved obtaining novel results inherent to the oxidation of the C12−C13 double bond for perezone and its synthetic isomer (isoperezone), involving different oxidation reactions and their corresponding physical properties, and spectroscopic characterization of the obtained products; furthermore, an in silico study for all the obtained molecules was performed at the DFT level employing the B3LYP functional, showing that the geometrical, spectroscopical, and reactivity properties agreed satisfactorily with the experimental results. Ultimately, molecular docking and a chemoinformatic prediction were also performed, which agreed conveniently with the cytotoxicity results.

## Data Availability

The data reported in this study are available upon request to mirruv@yahoo.com.mx.

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
