# Peer review of "Green Contributions to the Chemistry of Perezone and Oxidation of the Double Bond of the Side Chain: A Theoretical Study and Cytotoxic Evaluation in MDA-MB231 Cells"

_molecules, 2025, doi:10.3390/molecules30234603_

Round 1
Reviewer 1 Report
Comments and Suggestions for Authors
The present manuscript by René Gerardo Escobedo-González et al describes the preparation of eight compounds (i.e. four oxidation derivatives of perezone and four of isoperezone). These compounds were evaluated in vitro for their cytotoxicity against MDA-MB231 cells. The derivatives were also studied in silico (structural and docking). The manuscript could be of interest to researchers in the field. However, a number of revisions are necessary:
1) In lines 131 and 410 it is stated “…the most active compound (3)…” and “…Bearing in mind, that 3 presented experimentally the highest biological activity, followed by 6,…”. However, this is wrong as the most active compound, based on molarity (which should be used for all comparisons of structurally deferent compounds), is compound 6. Thus, all the manuscript should be modified based on this fact.
2) The correct structure of 7 and 8, based on 1H-NMR, is most likely the C12-acetate and not the C13-acetate.
3) A positive control should be included in Table 1.
4) The cytotoxicity of isoperezone should be included in Table 1.
5) All the reported new derivatives could exist in two diastereomeric configurations (8R-12R & 8R-12S). Were the studied compounds pure diastereomers (which ones?) or mixtures? This should be studied/reported.
6) The cytotoxicity of the compounds towards normal cells (especially the known for toxicity epoxides) should be studied/reported.
Reviewer 2 Report
Comments and Suggestions for Authors
The authors investigated the enzymatic oxidation of the sesquiterpene quinone perezone and its synthetic isomer isoperezone. They efficiently oxidized the C12–C13 double bond to produce epoxides, acetonides, diols, and esters, all characterized by spectroscopic methods. Among these, perezone epoxide exhibited the strongest cytotoxicity against breast cancer cells. DFT calculations revealed minimal energy differences (0.4 kcal/mol) between stereoisomers, consistent with experimental data. Molecular docking studies indicated strong binding affinity to His121 (−7.29 kcal/mol).
However, several revisions are required before the manuscript can be considered for publication:
- Please clarify the rationale for adopting a green synthesis approach instead of conventional oxidation methods.
- Do the authors believe that the use of Novozyme 435 contributed to the observed selectivity?
- The description of the synthesis from lines 88–95 (“In the second step, … generating the acetonides [9].”) is unclear and should be rewritten with greater detail and clarity.
- The statement “it is a yellow solid displaying a mp 78–80 °C, implying a pure compound” should be revised (line 112). The solid nature or melting point alone is not sufficient evidence of purity.
- Overall, the synthetic chemistry results and discussion section requires thorough revision.
Round 2
Reviewer 1 Report
Comments and Suggestions for Authors
The authors René Gerardo Escobedo-González et al did not answer at all my most crucial comment (i.e., comment 1). In this comment I pointed out to the authors that the most active compound, based on molarity (which should be used for all comparisons of structurally deferent compounds), is compound 6. The authors should appropriately revise their manuscript based on this comment.
Reviewer 2 Report
Comments and Suggestions for Authors
Accept in
